# Concept of an artificial muscle design on polypyrrole nanofiber scaffolds

**Madis Harjo[1], Martin Järvekülg[2], Tarmo Tamm[1], Toribio F. Otero[3], Rudolf Kiefer[4]\***

**1** Intelligent Materials and Systems Lab, Faculty of Science and Technology, University of Tartu, Tartu, Estonia, **2** Institute of Physics, Faculty of Science and Technology, University of Tartu, Tartu, Estonia, **3** Centre for Electrochemistry and Intelligent Materials (CEMI), Universidad Politécnica de Cartagena, Cartagena, Murcia, Spain, **4** Conducting polymers in composites and applications Research Group, Faculty of Applied Sciences, Ton Duc Thang University, Ho Chi Minh City, Vietnam

\* rudolf.kiefer@tdtu.edu.vn

**Data Availability Statement:** All relevant data are within the paper and its Supporting Information files.

**Funding:** The work was supported by Estonian Research Council Grant PRG772 The fund refer to Martin Järvekülg The funders had no role in study design, data collection and analysis, decision to publish, or preparation of the manuscript.

**Competing interests:** The authors have declared that no competing interests exist.

## Abstract

Here we present the synthesis and characterization of two new conducting materials having a high electro-chemo-mechanical activity for possible applications as artificial muscles or soft smart actuators in biomimetic structures. Glucose-gelatin nanofiber scaffolds (CFS) were coated with polypyrrole (PPy) first by chemical polymerization followed by electro-chemical polymerization doped with dodecylbenzensulfonate (DBS$^-$) forming CFS-PPy/DBS films, or with trifluoromethanesulfonate (CF$_3$SO$_3^-$, TF) giving CFS-PPy/TF films. The composition, electronic and ionic conductivity of the materials were determined using different techniques. The electro-chemo-mechanical characterization of the films was carried out by cyclic voltammetry and square wave potential steps in bis(trifluoromethane)sulfonimide lithium solutions of propylene carbonate (LiTFSI-PC). Linear actuation of the CFS-PPy/DBS material exhibited 20% of strain variation with a stress of 0.14 MPa, rather similar to skeletal muscles. After 1000 cycles, the creeping effect was as low as 0,2% having a good long-term stability showing a strain variation per cycle of -1.8% (after 1000 cycles). Those material properties are excellent for future technological applications as artificial muscles, batteries, smart membranes, and so on.

## 1. Introduction

The development of reliable soft robotics and smart devices mimicking the multifunctionality of natural organs of living creatures requires new actuators in biomimetic structures and artificial muscles, working at low potentials with low energy consumption while exhibiting high strain variation per potential cycle and high long-term stability. Conducting polymers actuators [1] that work at low voltage, undergo large bending displacements [2] and linear strains as high as 26% [3]; 20% of strain variation was reported in case of PPy-Platinum/Iridium coil structures [4]. Inspired by similar coil designs, researchers try to increase linear strain by using carbon nanotubes and other natural or artificial electrospun materials [5] in their plain form or with a conducting polymer coating (CP). In addition, the electrochemical reaction of the CP driving the actuator senses, simultaneously, the physical and chemical conditions [6]

allowing the development of sensing [7] and tactile [8] actuators suitable for mimicking the multifunctionality of haptic natural muscles and developing artificial proprioceptive systems [9,10]. At present, the main technological problem is the described poor long-term stability, the high strain variations are only achieved in the initial potential cycles.

The actuation mechanism has been intensely studied in the past [11–15] with no unified model established. The main idea behind any model or development is making an "artificial muscle" in biomimetic structures comparable to skeletal muscles [1] showing ~ 20% of strain variation during an actuation cycle developing a stress higher than 0.1 MPa (0.3 MPa from cardiac muscles). Natural muscles have fiber bundle structures constituted by chemical molecular machines, i.e. actin-myosin-ATP, that develop that strain and stress variation at high rates during milli seconds, comparable performance has not been achieved by conducting polymer-based actuators.

Fiber-actuators based on polyaniline have reached strains in range of 1.2% [16]. 3% strain has been reported for actuators based on interpenetrated networks of hollow fibers- of poly (3,4-ethylenedioxythiophene) [17]. Use of solid polymer electrolytes (SPE) [18] in linear fiber-actuators as well as encapsulated design [19] have led to linear strain of 0.5% [20] with draw-backs in actuator performance [21]. Different fiber-actuator designs of knitted and twisted yarns have been applied [22] to increase linear strain up to 3%. Using a single Lyocell PPy fiber a diametrical strain up to 6% was achieved [23]. Previous research [24] has also described a solution based on different elastic stretchable silicon yarns coated with electropolymerized PPy that reached 0.1% strain. Nanofiber scaffolds produced by electrospinning have fiber bundle structures that can be applied as is, or coated with PPy, in tissue engineering for skeleton muscle repair [25] due to its biocompatibility with cells [26,27]. Thus, besides the right conducting polymer, the substrate material has a great influence on the attained electrochemical actuation.

Following those ideas for increasing the mechanical performance of linear actuators our goal here was to use conductive nanofiber scaffold material coated with electropolymerized PPy for designing biomimetic active materials, which could be comparable to natural skeletal muscles in terms of structure, achievable strain and stress, and importantly—with long life time. The conductive nanofiber scaffold (CFS) consist of chemical coated used in previous works [28,29] as the substrate for electrodes to develop new materials by electropolymerization. This material can be stretched up to 17% without a major loss of conductivity.

## 1.1 Driving electrochemical reactions

Two different materials were obtained by using the conductive fiber scaffold (CFS) to electro-deposit a new PPy coat from solutions containing two different electrolytes: CFS-PPy/DBS and CFS-PPy/TF. Linear strain and force measurements were performed by cyclic voltammetry in LiTFSI propylene carbonate (PC) solutions. The material actuation (reversible volume change by swelling/shrinking) is driven by reversible electrochemical oxidation/reduction reactions of the PPy material exchanging anions and solvent with the electrolyte for charge and osmotic balance [30–32] following Eq 1: forwards, oxidation-swelling reaction; backwards, reduction-shrinking reaction:

$$[(PPy)\,(DBS^-)_m(Li^+)_m(PC)_o] + nTFSI^- + pPC \rightleftarrows$$
$$[(PPy)^{n+}(DBS^-)_m(Li^+)_m(PC)_{(o+p)}(TFSI^-)_n + n(e^-)_{metal}$$
(1)

The un-dissociated $Li^+DBS^-$ ions pairs were incorporated during the electropolymerization process remaining now trapped in PC solutions, as was proved by XPS. The PPy/TF material

also follows an anion driven actuation [33] without trapping ions seen in Eq 2:

$$(PPy^0) + n(TFSI^-)_{sol} + m(PC) \rightleftarrows [(PPy^{n+})(TFSI^-)_n(PC)_m] + n(e^-) \qquad (2)$$

The left side shows the reduced PPy material and the right side the oxidized PPy material where the solvated anions TFSI$^-$ incorporate in the positively charged PPy and are remove as the PPy is reduced. As a typical anion driven actuator, the simplest material model of the intracellular matrix (ICM) of the muscular cells [34].

## 2. Material and methods

### 2.1 Materials

Propylene carbonate (PC, 99%), ethanol (technical grade), sodium dodecylbenzene sulfonate (NaDBS, technical grade), ammonium persulfate (APS, technical grade) and tetrabutylammonium triflouromethanesulfonate (TBACF$_3$SO$_3$, 99%) were obtained from Sigma-Aldrich and used as supplied. Solvent ethylene glycol (EG, 99.8%) was purchased from Fluka. Pyrrole (Py, 98%, Sigma-Aldrich) was vacuum-distilled prior use and stored at low temperature in the dark. Milli-Q+ water was used for making aqueous solutions. Lithium bis(trifluoromethanesulfonyl)imide (LiTFSI, 99.95%) was purchased from Solvionic. Gelatin type A from porcine skin, D-(+)-glucose (99.5%) and glacial acetic acid (99%) were purchased from Sigma-Aldrich.

### 2.2 Electropolymerization of PPy films on conductive fiber scaffolds (CFS)

Fiber scaffolds based on type A gelatin from porcine skin (Sigma-Aldrich) and glucose were prepared by electrospinning, as previously described [28]. The fiber scaffolds were coated with a conductive polypyrrole material (thickness 16 ± 1.1 μm) to achieve a 49 ± 3 μm thickness of the conductive fiber scaffold, with electronic conductivity of 0.35 ± 0.02 S cm$^{-1}$. The attained CFS material was used as the working electrode for a subsequent galvanostatic electropolymerization (0.1 mA cm$^{-2}$, 40,000s, -40˚C) of polypyrrole from different polymerization solution. A two-electrode electrochemical cell was used. The working electrode—the CFS material (18 cm$^2$)—was placed in between two counter-electrodes of stainless steel mesh (each 18 cm$^2$), at ~ 1 cm distance from either.

The CFS-PPy/DBS samples were obtained using 0.1 M pyrrole and 0.1 M NaDBS ethylene glycol (EG) Milli-Q water (1:1) solutions. The samples, now coated with PPy-DBS, CFS-PPy/DBS, were washed with ethanol in order to remove any unpolymerized pyrrole and then with Milli-Q water in order to remove any excess of NaDBS. Then the samples were dried in the oven at 60˚C (2 mbar) for 12h. The final thickness of the CFS-PPy/DBS samples was 125 ± 8 μm. The CFS-PPy/TF samples were prepared by galvanostatic polymerization in 0.1 M pyrrole, 0.1 M TBACF$_3$SO$_3$ (TF) propylene carbonate solutions during a polymerization time of 40,000s. The coated CFS-PPy/TF samples were washed with propylene carbonate first in order to remove any excess of TBACF$_3$SO$_3$, then with ethanol to remove any excess of pyrrole monomer and then dried in the oven at 60˚C (2 mbar) for 12h. The final thickness of the dry CFS-PPy/TF samples was 136 ± 9 μm. For comparison, PPy/DBS and PPy/TF films were polymerized on stainless steel sheets using the same experimental condition and then removed from the metal to be used as free standing film electrodes. The thickness of the free-standing films was: 24 ± 2 μm for PPy/DBS and 20 ± 1 μm for PPy/TF.

### 2.3 Linear actuation

The CFS-PPy/DBS and CFS-PPy/TF samples and PPy/DBS and PPy/TF free standing films were cut into longitudinal strips: 1.0 cm length and 0.1 cm width each. In a three electrode set

up: platinum counter electrode, Ag/AgCl (3M KCl) reference electrode, the samples were fixed as the working electrodes on gold contact/electrode of a force sensor (TRI202PAD, Panlab) of the linear muscle analyzer setup [35]. The samples were pre-stretched by 2%, holding this position for 2h. Then the electro-chemo-mechanical experiments were carried out in 0.2 M LiTFSI in propylene carbonate (LiTFSI-PC) solutions by following: the sample length change under a constant force of 4.9 mN, or the force change for a constant length of 1 mm while submitting the sample to cyclic voltammetry (CV) at 5 mVs$^{-1}$, between 1.0 V and -0.55 V. The PPy/DBS and PPy/TF free standing films were submitted to the same electro-chemo-mechanical experiments. In order to explore the frequency-response and the useful life-time, the CFS-PPy/DBS and CFS-PPy/TF samples were submitted to consecutive square wave potential steps between the same potential limits in the frequency range between 0.0025 Hz and 0.1 Hz. Long term measurements (1000 cycles) were performed at 0.1 Hz. Three different samples for each synthesized material were measured. The presented values represent means, reported together with standard deviations.

The diffusion coefficients of the exchanged counter ions were calculated following the Eqs 3 and 4 [36].

$$ln\left[1 - \frac{Q}{Q_t}\right] = -bt \qquad (3)$$

$$D = \frac{bh^2}{2} \qquad (4)$$

From the current density/time experimental responses at each applied frequency, the charge density Q at each oxidation or reduction time point t, was determined by integration of the response, with $Q_t$ the charge density at the end of each reaction. By plotting (Eq 4) ln [1-Q/$Q_t$] versus the reaction time t, the slope, b, can be obtained. The diffusion coefficient D was determined from Eq 5 by including the thickness, h, of the sample.

## 2.4 Characterization of CFS-PPy/DBS and CFS-PPy/TF

The CFS-PPy/DBS and CFS-PPy/TF samples were characterized by scanning electron microscopy (Helios NanoLab 600, FEI) and energy-dispersive X-ray spectroscopy (EDX) (Oxford Instruments with X-Max 50 mm$^2$ detector). Before characterization, 5 min polarization at -0.55 V and +1.0 V was performed for the reduced and oxidized films, respectively. FTIR spectroscopy (Bruker Alpha with Platinum ATR) was applied to characterize CFS, CFS-PPy/DBS and CFS-PPy/TF samples in range of 2000 to 800 cm$^{-1}$. To evaluate the surface conductivity of the coatings, an in-house 4-point probe was used. The conductivity was calculated using the Eq 5:

$$\sigma_e = \frac{1}{(R * w)} \qquad (5)$$

where $\sigma_e$ is the electric conductivity, R is the surface resistivity ($\Omega$/sq) and w is the material thickness.

## 3. Results and discussion

### 3.1 Electropolymerization and morphology

The electropolymerization curves (chronopotentiometric responses during the electropolymerization process) and the SEM images of the attained free standing PPy/DBS and PPy/TF films are presented in S1 Fig. The electropolymerization responses during the synthesis of the

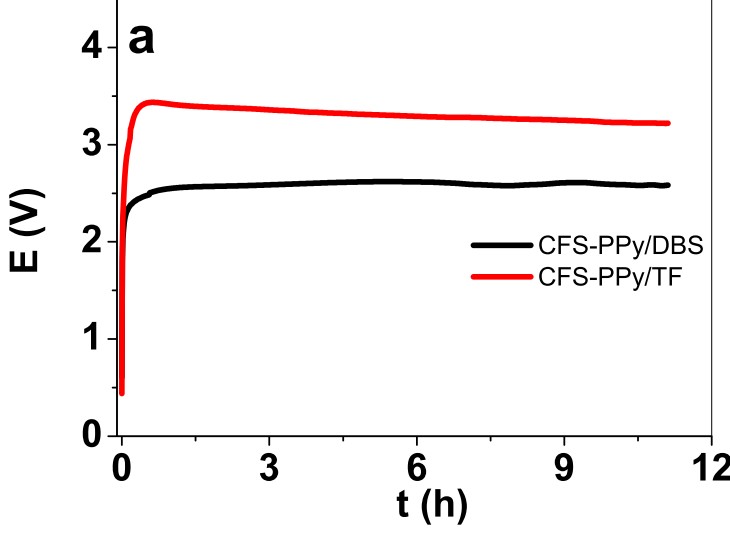

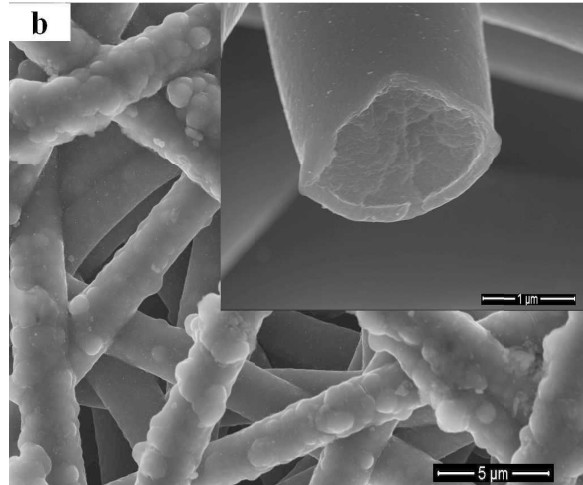

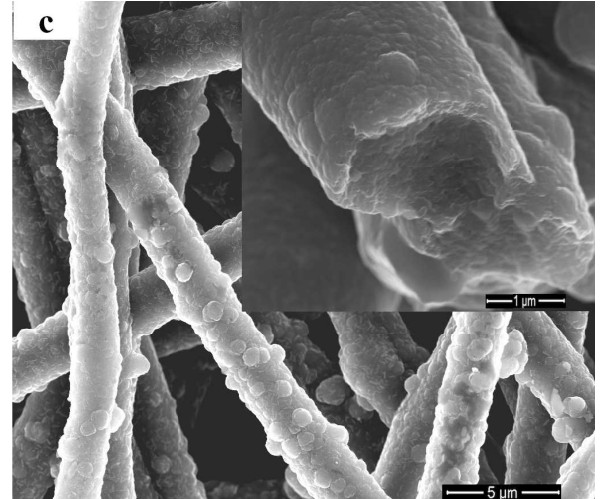

**Fig 1.** a: Chronopotentiometric response during the galvanostatic electropolymerization of the CFS-PPy/DBS (black line) and CFS-PPy/TF (red line) scaffold materials. The SEM surface images (scale bar 5 μm); each inset shows the cross section of a single fiber (scale bar 1 μm): b, CFS-PPy/DBS, and c, CFS-PPy/TF.

CFS-PPy/DBS and CFS-PPy/TF films, as well as the SEM images of the attained materials are shown in Fig 1A–1C.

Fig 1A shows that the CFS-PPy/DBS samples grew during 11.1h under a constant electro-polymerization potential of 2.55 V, while the electropolymerization of the CFS-PPy/TF sample was a more resistive process, going through a potential maximum of 3.45 V after 0.5h and then decreasing gradually to 3.22 V by the end of the polymerization time. These observations can be attributed to the different electronic conductivity of the samples immersed in different electrolytes; $0.42 \pm 0.2$ S cm$^{-1}$ in aqueous-ethylene glycol solution, and $0.11 \pm 0.1$ S cm$^{-1}$ in PC solution similar to those found in a previous work [29]. The reference electropolymerization of PPy/DBS and PPy/TF films on stainless steel sheets (S1 Fig) required lower voltages: 1.17 V for PPy/TF and 0.9 V for PPy/DBS, due to the higher conductivity of the underlayer, but also pointing to less overoxidation-degradation processes during the material generation. Fig 1B shows that the CFS-PPy/DBS sample presents a quite uniform coating film of electrodeposited

PPy/DBS material. The diameter of the cross section of the fiber is 2.2 ± 0.2 μm. The diameter of the original chemically coated nanofiber scaffold was 1.4 ± 0.1 μm [28], accordingly, the PPy/DBS electrodeposition added 0.8 ± 0.04 μm (inset of Fig 1B). The CFS-PPy/TF sample (inset of Fig 1C) shows a diameter in range of 2.5 μm. The SEM images of PPy/DBS and PPy/TF free standing films, S1B and S1C Fig, revealed the typical PPy morphology seen before [37,38], with a dense cross section (inset of S1B and S1C Fig). Table 1 present both, conductivities and thicknesses of the dry samples, as well as those of the free standing PPy/DBS and PPy/TF films.

The free standing films present the highest conductivities, as expected for solid films, 15.4 S cm$^{-1}$ for PPy/DBS and 11.4 S cm$^{-1}$ for PPy/TF quite close to the CFS-PPy/TF sample, 10 S cm$^{-1}$; the CFS-PPy/DBS sample was the most resistive, at 2.4 S cm$^{-1}$. The results can be related to the influence of the PC solvent on the electropolymerization process [39], but also on the different real (active material) density of the materials.

**3.1.1 FTIR and EDX spectroscopy.** Fig 2A shows the FTIR spectra obtained from the original CFS material and from the CFS-PPy/DBS and CFS-PPy/TF samples. Fig 2B and 2C show the EDX spectroscopic results (using cross section of the CFS-PPy/DBS and CFS-PPy/TF) from both, the oxidized and the reduced samples in order to try to identify the different ion content of the two states.

The CFS material (Fig 2A) contained some PPy from the original chemical polymerization (using APS oxidant) on the glucose-gelatin nanofiber scaffold. The sharp peak at 1778 cm$^{-1}$ seen in CFS sample is from the carboxyl group (C = O) formed by PPy overoxidation [40]. Typical signals for glucose-gelatin are shown at 1640 and 1533 cm$^{-1}$ [41] as the amide I and amide II peaks, respectively. The peak at 1640 cm$^{-1}$ can be clearly identified in CFS, CFS-PPy/TF and CFS-PPy/DBS. The peak at 1533 cm$^{-1}$ for CFS-PPy/DBS is overlapped by the 1542 cm$^{-1}$ peak, which represents the polypyrrole ring stretching vibration (C = C) [42]. Other typical PPy peaks can be found at 1452 cm$^{-1}$ (C-C stretching vibrations) and 1280 cm$^{-1}$ (C-N stretching mode) [43] and the band at 1160 cm$^{-1}$–1200 cm$^{-1}$, which was identified in CFS-PPy/DBS at 1168 cm$^{-1}$ as the breathing vibration of the PPy ring [44]. The 1183 cm$^{-1}$ peak found for CFS and CFS-PPy/TF describe the same breathing vibrations of PPy rings [44]. The polaron band is found in CFS and CFS-PPy/DBS at 1040 cm$^{-1}$ (in literature 1045 cm$^{-1}$ [45]). In CFS-PPy/TF a new peak at 1023 cm$^{-1}$ was associated with the triflate anion [46]. It can be concluded that PPy/DBS and PPy/TF were indeed deposited on CFS material.

Fig 2B and 2C show the EDX spectroscopic results from CFS-PPy/DBS and CFS-PPy/TF samples, respectively, in the oxidized (at 1.0 V) and the reduced (at—0.55 V) states. The peaks correspond to the different atoms present in the samples: carbon (C) at 0.27 keV, nitrogen (N) at 0.38 keV, oxygen (O) at 0.52 keV, fluorine at 0.68 keV and sulfur (S) at 2.32 keV. The oxidized CFS-PPy/DBS material presented an increase of fluorine, oxygen and sulfur content compared to those of the reduced material indicating the incorporation of TFSI$^-$ anions during the oxidation reaction. A small amount TFSI$^-$ remained trapped in the reduced state of the CFS-PPy/DBS material, as observed before [38] using free standing PPy/DBS films. The high amount of sulfur after reduction refers to: entrapped TFSI$^-$ anions, the immobile DBS$^-$ anions

**Table 1. Electronic conductivities and thicknesses of CFS-PPy/DBS and CFS-PPy/TF samples as well PPy/DBS and PPy/TF free standing films (after polymerization).**

| Samples | Conductivity [S cm$^{-1}$] | Thickness [μm] |
|---|---|---|
| CFS-PPy/DBS | 2.4 ± 0.2 | 125 ± 8 |
| PPy/DBS | 15.4 ± 0.5 | 24 ± 1.1 |
| CFS-PPy/TF | 10.0 ± 0.7 | 136 ± 9 |
| PPy/TF | 11.4 ± 0.8 | 20 ± 0.1 |

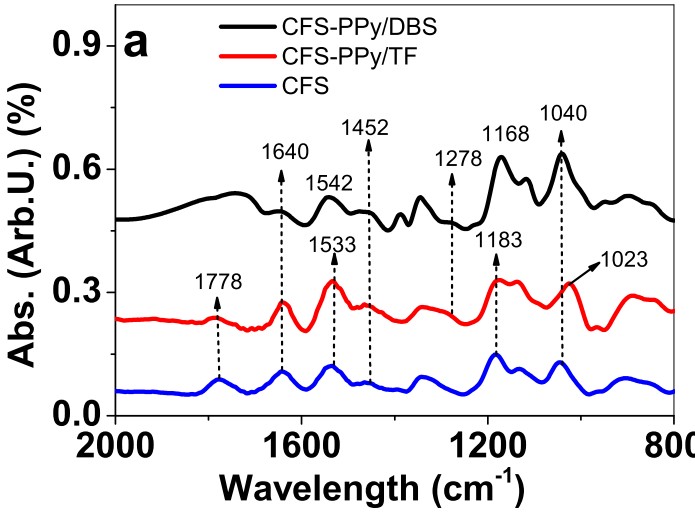

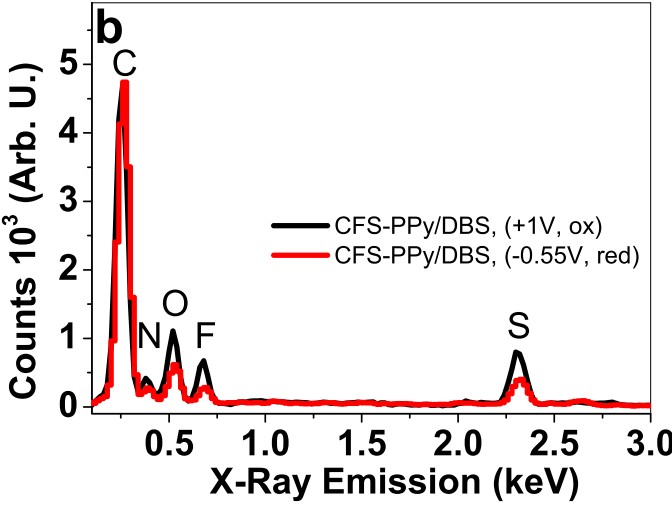

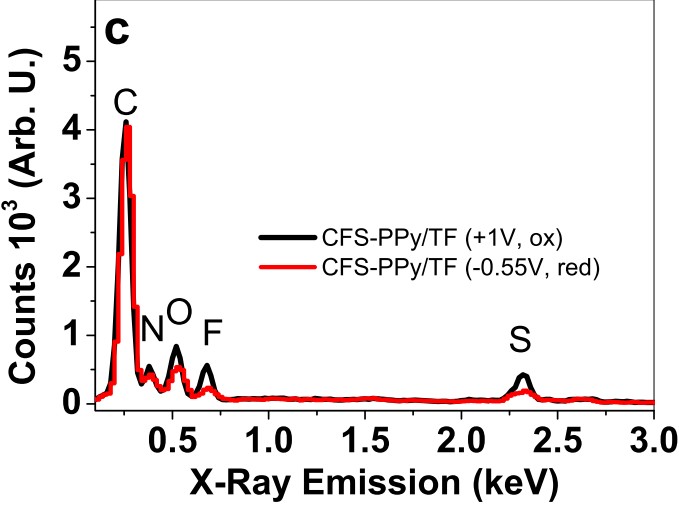

**Fig 2.** a: FTIR spectra of CFS-PPy/DBS (black line), CFS-PPy/TF (red line) and CFS samples (blue line) in wavelength range of 2000 to 800 cm$^{-1}$. EDX spectra of oxidized (5min, 1V, black line) and reduced (5 min at -0.55 V, red line) in b: CFS-PPy/DBS and c: CFS-PPy/TF.

in PPy and some $SO_4^{2-}$ anions from the initial chemical polymerization of PPy on CFS [47]. The peaks of oxygen, fluorine and sulfur increased after the oxidation of the CFS-PPy/TF samples, indicating again the entrance of balancing TFSI$^-$ anions from the solution during the oxidation of the material. After reduction, some fluorine, related to the residual immobilized spherical triflate anions [48] ($CF_3SO_3^-$) from the PPy electropolymerization were present, as also seen previously [37]. As a partial conclusion, the EDX results indicate that the CFS-PPy/DBS and CFS-PPy/TF samples follow anion-active TFSI$^-$ exchange for charge balance during the reversible redox cycles.

### 3.2 Linear actuation

The bundle-like fibers coated individually with PPy (as seen in Fig 1B and 1C) are envisaged to bring closer the concept of an "artificial muscle". To mimic the natural contraction/elongation

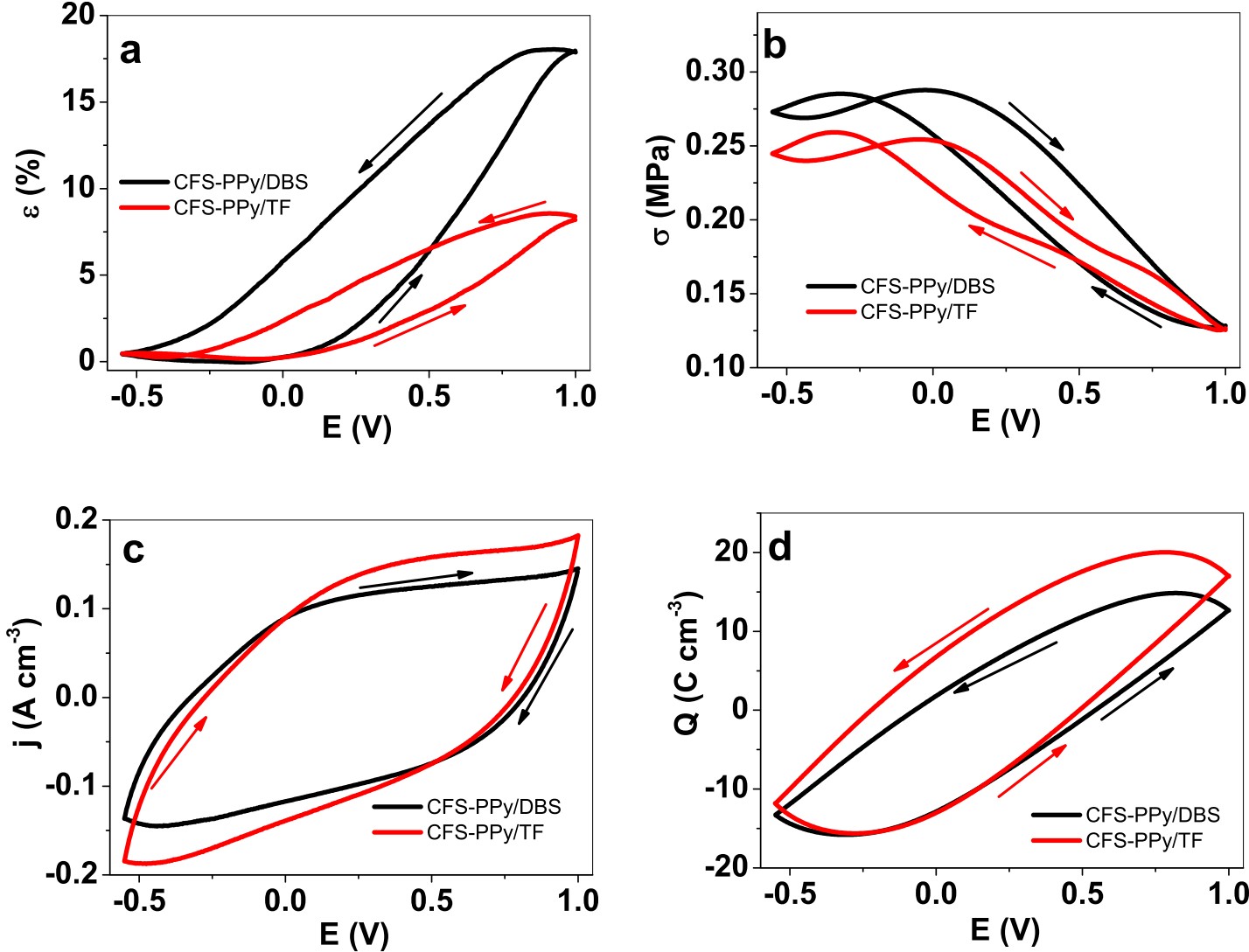

**Fig 3.** Cyclic voltammetry (scan rate 5 mV s$^{-1}$) of CFS-PPy/DBS (black line) and CFS-PPy/TF (red line) in LiTFSI-PC (potential range 1 V to -0.55 V against Ag/AgCl (3 M KCl) reference electrode, 4$^{th}$ cycle) showing in a: strain ε; b: stress σ, c: current density j and d: charge density Q against potential E. The arrows indicate the direction of scan.

of muscle tissue, linear actuation of the CFS-PPy/DBS and CFS-PPy/TF samples was carried out to ensure that those materials could be suitable for the envisaged soft robots and smart multifunctional applications. The aforementioned suitability was further investigated by long-term measurements, at 0.1 Hz frequency to prove the durability of the material and consistency of the response.

**3.2.1 Reversible electro-chemo-mechanical changes under cyclic voltammetry.** Cyclic voltammetry results describe the response of the materials to closer-to-equilibrium conditions, as the driving signal is a (relatively slow) sweep rather than the abrupt polarity change of square wave potential steps. Fig 3 presents the CV (scan rate 5 mV s$^{-1}$) driven results, with the electro-chemo-mechanical response of strain and stress in 3a and 3b, respectively. The voltammetric responses (current density vs. potential) are shown in Fig 3C. Fig 3D presents the charge density (mC cm$^{-3}$) calculated by the integration of the cyclic voltammograms, taking

into account the sweep rate. Fig 3A reveals that the samples swell/shrink by oxidation/reduction, respectively, indicating anion-driven actuation (Eq 1 for PPy/DBS and Eq 2 for PPy/TF materials). Upon oxidation, the anions ingress into the material from the solution in order to balance the positive charges generated by electron loss on the polymer chains. The material swelling results, under constant force, in the increase of strain, up to 18% for CFS-PPy/DBS and 8.2% for CFS-PPy/TF films. Upon reduction, the material recovers the original position of the beginning of the potential cycle. To our knowledge, the results presented here represent the highest strain variation reported for any nanofiber scaffold materials.

The stress variations at constant length, Fig 3B, reveal a similar response for both the studied materials: 0.14 MPa for CFS-PPy/DBS and 0.12 MPa for CFS-PPy/TF, both in the range of the skeletal muscles (0.1 MPa). The reference actuator responses from free-standing films, S2A Fig, show a reversible strain variation in range of 20% for the PPy/DBS, and 8.6% for the PPy/TF films. The stress evolves, S2B Fig, in range of 0.24 MPa for PPy/DBS and 1.38 MPa for PPy/TF free standing films. The presence of a stress maximum and two crossing loops suggest the existence of some mixed ion actuation (entrance of some cations at the most cathodic potentials). Such excellent strain performance could be attributed to the underlying CFS material, as indicated in previous works [28]. Typically, conducting polymers have been deposited on inert metals such as (sputtered) gold or platinum, which can develop failures by cracking if stretched [49], reducing the conductivity up to 25 times after 20% stretching, becoming non-conductors after 30% stretching. Recently [29] it was observed that the conductivity of the CFS in aqueous solution under modified EIS measurements dropped by 50% after 16,7% stretching.

The closed charge density loops in Fig 3D indicate full reversibility of the oxidation/reduction reactions inside the studied potential range: the presence of irreversible parallel oxidation or reduction reactions, i.e., solvent electrolysis, should give open Q/E loops. This confirms that the anodic/cathodic charges are used to oxidize/reduce the conducting polymer ("steady state"). The reversible redox cycles are expected to result in faradaic actuators [50]: linear variation of the strain with the reaction charge density. For comparison, the cyclic voltammetric results attained from free standing PPy/DBS and PPy/TF films are shown in S2C Fig and those for CFS-PPy/DBS and CFS-PPy/TF samples in Fig 3C. The maximum redox charge (Fig 3D and S2D Fig) gives the involved charge densities as: 30.4 C cm$^{-3}$ for CFS-PPy/DBS, 42 C cm$^{-3}$ for CFS-PPy/TF, 90.4 C cm$^{-1}$ for PPy/DBS and 78.5 C cm$^{-1}$ for PPy/TF. The higher densities for free-standing films are logically explained by their much higher content of active material.

The Young's modulus of conducting polymers also typically changes during reversible redox cycles, altering the response, as it becomes lower in the reduced than in the oxidized state [51]. Continues cycling/actuation also tends to cause changes. Table 2 shows the Young's modulus attained for our oxidized materials before and after actuation cycles.

The high value of the modulus of the PPy/DBS and PPy/TF free standing films is attributed to the significantly denser and compact structure. In addition to the obviously more fibrous structure of the CFS, the conducting polymer deposited on the CFS samples is also less compact, as the synthesis potential was higher. Whichever the studied sample, the modulus

**Table 2. Young's Modulus of linear actuator samples in LiTFSI-PC before and after actuation cycles.**

| Samples | Young's Modulus [kPa] | Young's Modulus [kPa] |
|---|---|---|
| | Before actuation (oxidized state) | After actuation (oxidized state) |
| PPy/DBS | 980 ± 60.3 | 700.2 ± 50.4 |
| PPy/TF | 260 ± 16.5 | 222.4 ± 12.3 |
| CFS-PPy/DBS | 123.8 ± 9.8 | 22.7 ± 1.2 |
| CFS-PPy/TF | 55.5 ± 4.7 | 43.2 ± 3.3 |

decreases after cycling in LiTFSI-PC (Table 2). This drop can be related [52] to the exchange of solvent and ions, creating a dense PPy gel and the modulus decreases [53]. For CFS-PPy/DBS samples the modulus decreases up to 5 times after actuation, having a direct (positive) effect on strain, as previously found for PPy-CDC or PPy-PEO/DBS free standing films [54,55]. Otherwise, the PPy/DBS films operating in LiTFSI-PC (Eq 1) contained un-dissociated Li⁺DBS⁻ ion couples [38] that tend to precipitate, accordingly, the exchanged counterions increase the amount of solvent in the film to maintain the osmotic balance [32] during the redox cycles, thus, decreasing the modulus.

**3.2.2 Square wave potentials steps and actuator life-time.** Three different samples of every studied material were submitted to consecutive square potential cycles involving a high oxidation/reduction charge density at different frequencies ranging between 0.0025 Hz and 0.1 Hz in order to follow the strain evolution during the actuator life-time (S1 Table). Fig 4A

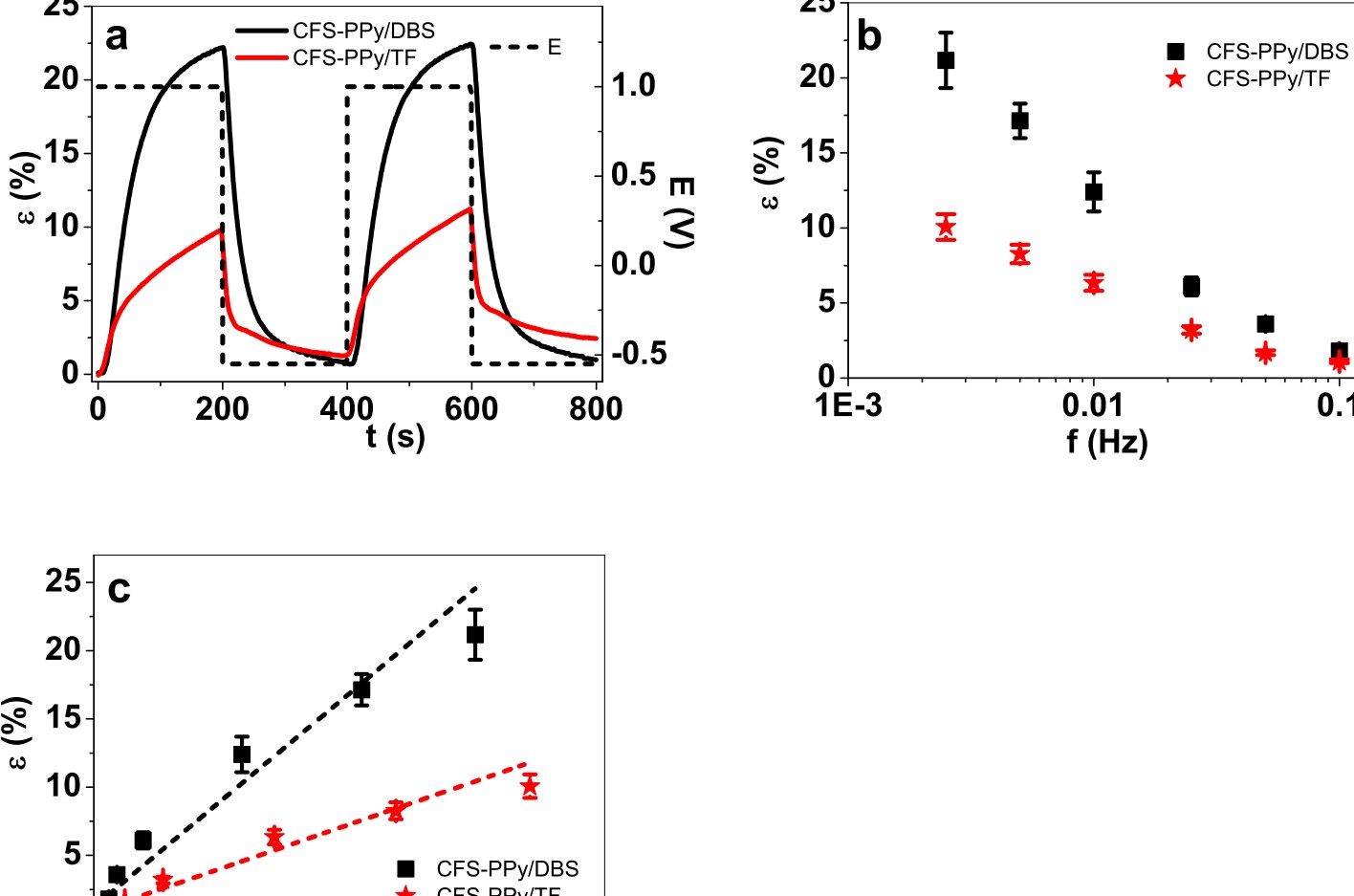

**Fig 4.** Square wave potential measurements in potential range 1 V to -0.55 V in LiTFSI-PC electrolyte showing in a: the strain ε (2nd-3th cycles) of CFS-PPy/DBS (black line) and CFS-PPy/TF (red line) with potential E (dashed) against time t (frequency, 0.0025 Hz). At frequencies 0.0025 Hz to 0.1 Hz, the maximum strain of CFS-PPy/DBS (black, ■) and CFS-PPy/TF (red, ★) against logarithmic scale of frequency are shown in (b). The obtained charge densities Q from each chronopotentiogram shown in c: strain ε against the charge density Q. The dashed lines in c represent linear fits and are shown just as visual guides.

shows the applied potential cycles (dashed lines) and the corresponding strain response of the materials (CFS-PPy/DBS and CFS-PPy/TF). The mean strain values with standard deviations as a function of both, the applied frequency and the redox charge are shown in Fig 4B and 4C, respectively.

At the lowest frequency, 0.0025 Hz, corresponding to higher oxidation/reduction charge, 22% of strain variation was recorded (Fig 4A) for the CFS-PPy/DBS material. This high strain corroborates the above attained (18%) by cyclic voltammetry (Fig 3A). Creep (different material length at the beginning and at the end of the cycle) was noticeable after two consecutive cycles: 1.2% for the CFS-PPy/DBS samples and 2.5% for CFS-PPy/TF samples. Rising loads (MPa) and lower frequencies (longer time in oxidation/reduction) produce higher creeping effects [56]. Higher oxidation/reduction charges at lower frequencies with more ions exchanged per cycle increases the viscosity PPy[57] and more solvent exchange enhances the plasticization process [32], allowing increased creep. The presence of some parallel irreversible reactions can also increase the creeping effect [58].

Fig 4B and 4C present the strain evolution as a function of the applied frequency and of the redox charge density, respectively. As usual, the strain decreases with increasing frequency; however, at each frequency the strain shown by CFS-PPy/DBS was two times higher than for the CFS-PPy/TF. The charge densities were obtained at each applied frequency from the concomitant chronoamperometric (current/time) responses. The strain of every material increased linearly with the redox charge density (Fig 4C), as expected for any faradaic actuators [50], as the volume variation (exchange of ions and solvent) is controlled by the redox charge. Whichever the frequency, for the same redox charge density the strain variation of the CFS-PPy/DBS samples is over two times that of the CFS-PPy/TF samples. Clearly, the coupling between ion/solvent flux and polymer matrix is much stronger in the former, likely due to the structure generated by the original doping ions.

The interaction of ion flux with the polymer matrix upon redox cycling must reflect in the (apparent) diffusion coefficient of the counterions through the material. Thus, the diffusion coefficients during oxidation and reduction were calculated using Eqs 3 and 4, for CFS-PPy/TF and CFS-PPy/DBS samples with thickness of 136 μm and 125 μm, respectively.

Fig 5A shows the linear dependence of the diffusion coefficients during the oxidation reaction on the applied frequency. The diffusion coefficients upon reduction against the applied frequency are shown at S3 Fig. Fig 5B presents the linear relationships between the strain rate (% s$^{-1}$) and the diffusion coefficients during the material oxidation.

Both Figs agree to the electrochemically stimulated conformational-relaxation (ESCR) model [11]: at higher frequencies the average swelling/shrinking change driven by the conformational relaxation of the polymeric chains of the conducting polymer are slower giving a lower density gel structure, with concomitant higher diffusion coefficients. In parallel, higher strain rate $v_{ox}$ of CFS-PPy/DBS was attained with rising experimental frequencies which correspond to, as stated in the previous sentence, higher diffusion coefficients: a linear relationship is attained between the strain rate and D.

The suitability of those two materials (CFS-PPy/DBS and CFS-PPy/TF) in biomimetic structure with such high strain for applications in soft robotics, artificial muscles and multifunctional devices requires long term stability of the actuation response. Fig 6 presents the results of the strain during 1000 actuating cycles at 0.1 Hz (S3 Table).

The creep after 1000 cycles was (Fig 6) 0.2% for CFS-PPy/DBS samples and 0.7% for CFS-PPy/TF samples. Of the studied materials, CFS-PPy/DBS is better suited for most applications when the strain response as well as the creep effect are considered. The strain variation per cycle decreases with the number of cycles (Fig 6B): 1.83% after 50 cycles and 1.75% after 1000 cycles for CFS-PPy/DBS and 1.09% after 50 cycles and 0.95% after 1000 cycles for

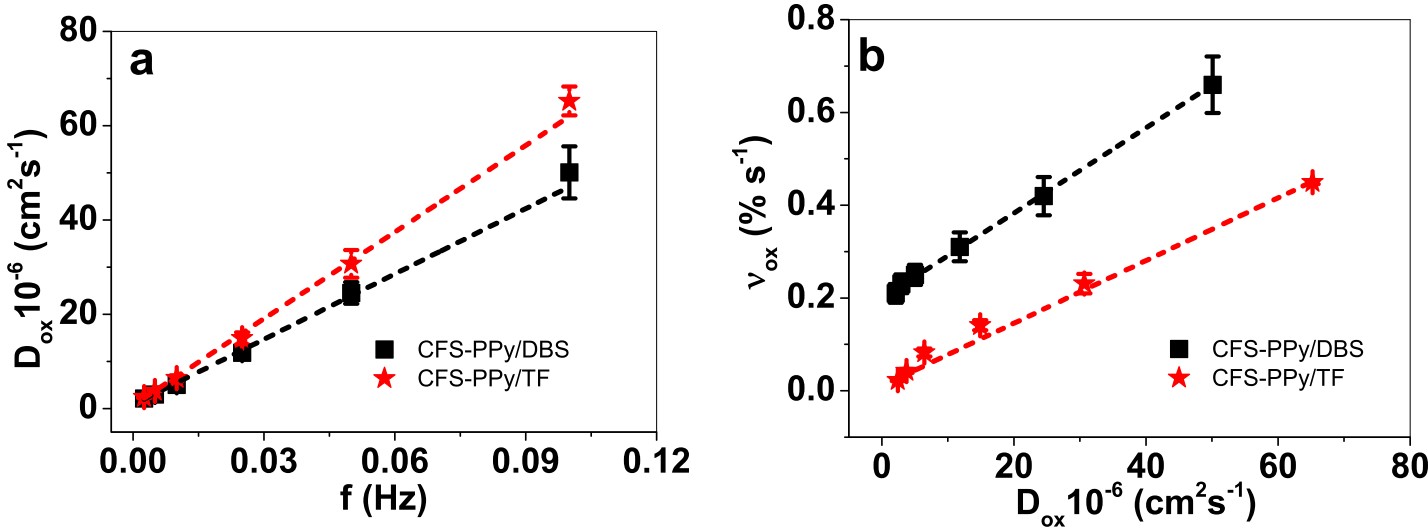

**Fig 5. CFS-PPy/DBS (■) and CFS-PPy/TF (★) showing in a: Diffusion coefficients $D_{ox}$ (upon oxidation) obtained from Eqs 2 and 3 against the applied frequency f and b: the strain rate $\nu_{ox}$ against diffusion coefficients $D_{ox}$.** The dashed lines represent the linear fit and shown for orientations.

CFS-PPy/TF. While the strain was higher for CFS-PPy/DBS, the loss of strain was lower for CFS-PPy/TF, indicating higher stability, which can be beneficial in technological applications.

Overall, the CFS-PPy/DBS material with a biomimetic structure presents great potential for technological applications in soft robotics applications due to its high strain of over 20%, low creeping effect (0,2% after 1000 cycles) and long term persistence of the strain variation achieved in a cycle (-1.8% after 1000 cycles). All of these compare favorably to natural skeletal muscles, although at a lower response rate. The actuation speed needs be optimized in future studies.

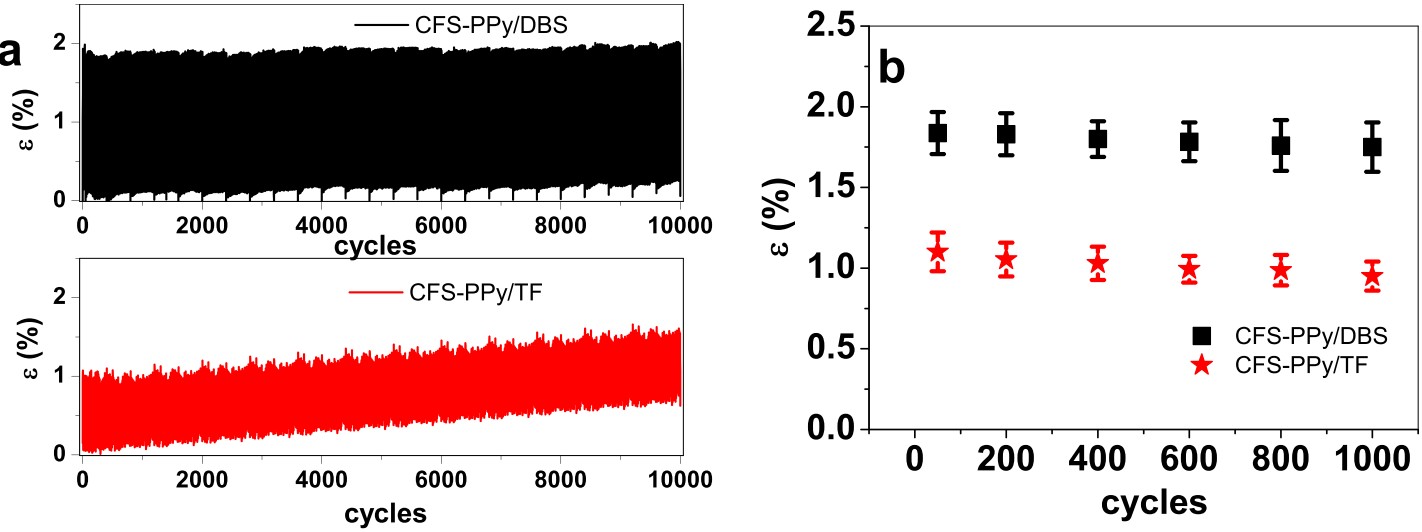

**Fig 6. Stability test: Square wave potential steps at frequency 0.1 Hz (1000 cycles) in potential range 1 V to -0.55 V in LiTFSI-PC showing in a: the strain ε of CFS-PPy/DBS (black) and CFS-PPy/TF (red) against time t and in b: the strain ε (main values with standard deviation) of CFS-PPy/DBS (■) and CFS-PPy/TF (★) against cycle number.**

## 4. Conclusions

We have shown that the CFS materials coated with conducting polymers (PPy/DBS and PPy/TF) remained with a biomimetic bundle-like structure (as seen in the SEM images). The FTIR measurements identified successful combination of CFS with PPy/DBS and PPy/TF. It was shown that 20% linear strain and 0.1 MPa stress for CFS-PPy/DBS (CFS-PPy/TF reached 8% linear strain) was achievable, which is similar to the respective parameters of natural skeletal muscles. For both materials, a single ion species (the anion) dominates the actuation direction (expansion at oxidation). Of the two materials, CFS-PPy/DBS should be preferred based on the maximum strain and stress reached, but also for lower creep with useful response stability in square wave potential step measurements over 1000 cycles at 0.1 Hz. The behavior is well described using theoretical models, allowing consistent characterization and control of potential devices. The main drawback of the fibrous linear actuators compared to natural muscles is the significantly lower strain rates. Further research in optimizing is needed in order to reach applications in soft robotics and multifunctional smart materials truly mimicking natural muscles.

## Supporting information

**S1 Fig. Free standing PPy/DBS and PPy/TF films electropolymerization and SEM images.**
(DOCX)

**S2 Fig. Cyclic voltammetry of PPy/DBS and PPy/TF free standing samples.**
(DOCX)

**S3 Fig. Diffusion coefficients at reduction for CFS-PPy/DBS and CFS-PPy/TF samples.**
(DOCX)

**S1 Table. Data of CFS-PPy/DBS and CFS-PPy/TF samples in square wave potential step measurements.**
(DOCX)

**S2 Table. Data for diffusion coefficients at oxidation and reductions including strain rates.**
(DOCX)

**S3 Table. Data for long term measurements of CFS-PPy/DBS and CFS-PPy/TF.**
(DOCX)

## Author Contributions

**Conceptualization:** Martin Järvekülg, Tarmo Tamm, Rudolf Kiefer.

**Data curation:** Madis Harjo, Martin Järvekülg, Tarmo Tamm.

**Formal analysis:** Madis Harjo.

**Funding acquisition:** Tarmo Tamm.

**Investigation:** Madis Harjo, Tarmo Tamm, Toribio F. Otero.

**Methodology:** Madis Harjo, Martin Järvekülg, Tarmo Tamm.

**Project administration:** Tarmo Tamm, Rudolf Kiefer.

**Supervision:** Martin Järvekülg, Rudolf Kiefer.

**Validation:** Toribio F. Otero.

**Writing – original draft:** Rudolf Kiefer.

**Writing – review & editing:** Martin Järvekülg, Tarmo Tamm, Toribio F. Otero, Rudolf Kiefer.

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
