## [Decision Letter · Decision Letter 0]

15 Apr 2020

PONE-D-20-03013

Concept of an artificial muscle design on polypyrrole nanofiber scaffolds

PLOS ONE

Dear Dr Rudolf Kiefer,

Thank you for submitting your manuscript to PLOS ONE. After careful consideration, we feel that it has merit but does not fully meet PLOS ONE’s publication criteria as it currently stands. Therefore, we invite you to submit a revised version of the manuscript that addresses the points raised during the review process.

We would appreciate receiving your revised manuscript by 17/04/2020. To enhance the reproducibility of your results, we recommend that if applicable you deposit your laboratory protocols in protocols.io, where a protocol can be assigned its own identifier (DOI) such that it can be cited independently in the future. For instructions see: http://journals.plos.org/plosone/s/submission-guidelines#loc-laboratory-protocols

We look forward to receiving your revised manuscript.

Kind regards,

Susana Fernandes

Academic Editor

PLOS ONE

Reviewers' comments:

Reviewer's Responses to Questions

**Comments to the Author**

1. Is the manuscript technically sound, and do the data support the conclusions?

Reviewer #1: Yes

Reviewer #2: Yes

Reviewer #3: Yes

2. Has the statistical analysis been performed appropriately and rigorously? 

Reviewer #1: No

Reviewer #2: Yes

Reviewer #3: Yes

3. Have the authors made all data underlying the findings in their manuscript fully available?

Reviewer #1: Yes

Reviewer #2: Yes

Reviewer #3: Yes

4. Is the manuscript presented in an intelligible fashion and written in standard English?

Reviewer #1: Yes

Reviewer #2: Yes

Reviewer #3: Yes

5. Review Comments to the Author

Reviewer #1: In this paper the authors report on experiments using fibre scaffolds.

It is not clear how many replicates the data are based upon an this should usefully be stated. Additionally, we have ambiguous error bars and use of the +/-: these can variously mean +/- sd; +/- s.e.m. or 95% CI for either data or the mean. All of these of course do depend on the number of data points contributing as well, either in terms of scale or accuracy. This could usefully be made clearer in the paper.

When describing something as changing, it is insufficient to rely solely on the numerical values - please apply a suitable test (presumably a paired one for before/after in the case of table 2).

How good is the fit of the linear model - given the potential for turning over (second derivative <0) in 4c, is a linear model justified mathematically (as opposed to statistically)?

Reviewer #2: In this manuscript is presented the synthesis and characterization of two novel conducting materials that have a high electro-chemo-mechanical activity based on glucose-gelatin nanofiber scaffolds coated with polypyrrole and doped with dodecylbenzensulfonate or with triflouromethanesulfonate. The experiments carried out in the manuscript are described in sufficient detail. The composition, electronic , ionic conductivity and the electro-chemo-mechanical activity was characterized by means of different characterization techniques. In my opinion, this manuscript is very interesting and the results, in addition to be well organized and discussed, they were compared and contrasted with literature data, being the conclusions presented supported by the reported data. Therefore, in my opinion, the manuscript is acceptable for publication as it is.

Reviewer #3: I have reviewed the research manuscript draft "Concept of an artificial muscle design on polypyrrole nanofiber scaffolds" written by Madis Harjo, Martin Järvekülg, Tarmo Tamm, Toribio F. Otero and Rudolf Kiefer submitted for publication in PLOS ONE.

After the reading of the manuscript, in my opinion it is an interesting work where the obtained results are reasonable well explained and could gives rise to a future research works in this line

Acceptable under minor revisions. Only I have few comments or suggestions.

1: Replace in the manuscript and supplementary information:

Madis Harjo1, Martin Järvekülg2, Tarmo Tamm1, Toribio F. Otero3, and Rudolf Kiefer4,*

by

Madis Harjo1, Martin Järvekülg2, Tarmo Tamm1, Toribio F. Otero3 and Rudolf Kiefer4*

2: In the abstract replace: triflouromethanesulfonate by trifluoromethanesulfonate

3: In the introduction, at the end of second paragraph: Comparable, without capital letter: comparable

4: Section 2.2: in the first (40ºC) and second paragraph (60ºC, 60ºC) replace de cero in superindex by degree symbol.

5: At the end of equation 3, 4 and 5, remove the comma, point and comma, respectively

6: Section 3.2.1, in first line of second paragraph: Figure 3b instead Figure 2b

7: Last line before section 3.2.2: “cycles ,”  “cycles,”

8: Section 3.2.2 in first paragraph: “CFS-PPy/DBS” instead of “PPy/DBS”

9: Section 3.2.2 in the paragraph prior to Figure 5:

Revise the sentence “Figure 5a shows the linear dependence of the diffusion coefficients during the oxidation reaction (S2 Table) on the applied frequency (S3 Figure) depicts the evolution during reduction reactions)”

10: Section 3.2.2 first paragraph after Figure 5: “sentence” instead of “sentiency”.

6. PLOS authors have the option to publish the peer review history of their article (what does this mean?). If published, this will include your full peer review and any attached files.

Reviewer #1: No

Reviewer #2: No

Reviewer #3: Yes: Asier Martinez Salaberria

---

## [Author Response · Author response to Decision Letter 0]

21 Apr 2020

Reviewer #1: In this paper the authors report on experiments using fibre scaffolds.

We thank the reviewer for the useful comments and have revised our manuscript regarding the suggestions

It is not clear how many replicates the data are based upon an this should usefully be stated. Additionally, we have ambiguous error bars and use of the +/-: these can variously mean +/- sd; +/- s.e.m. or 95% CI for either data or the mean. All of these of course do depend on the number of data points contributing as well, either in terms of scale or accuracy. This could usefully be made clearer in the paper.

Yes, a good point! All the reported numeric results are mean values of at least triplicate experiments, not just repeats from the same sample but different specimens. The +/- represent (single) standard deviations. This information was reported in the end of section 2.3 but it has been rewritten to improve clarity:

Three different samples for each synthesized material were measured. The presented values represent means, reported together with standard deviations.

When describing something as changing, it is insufficient to rely solely on the numerical values - please apply a suitable test (presumably a paired one for before/after in the case of table 2).

The material property change is a very important consideration in any polymeric or soft material in general, even more so with conducting polymers that are known to have elastic modulus change during redox processes. However, we are not quite sure what to improve here, since the values in table 2 are already exactly what the reviewer appears to be suggesting – values of the modulus before and after actuation experiments. Since the measurement of the elastic modulus is a dynamic process in itself (relation between the applied force/weight and deformation), it cannot be done concurrently with the actuation measurement, just before and after.

How good is the fit of the linear model - given the potential for turning over (second derivative <0) in 4c, is a linear model justified mathematically (as opposed to statistically)?

We have to totally agree with the reviewer. The data points in figure 4c do not follow a linear trend too well. As stated in the text (hopefully more clearly now in the revised version), the linear fits were just presented as visual guides, as in ideal case, the linear relation between charge and strain is expected. However, real systems are seldom ideal and the coupling between charge/ion flux and volume change (strain) changes with driving frequency, as variation in solvation and various relaxations come to play.

Reviewer #2: In this manuscript is presented the synthesis and characterization of two novel conducting materials that have a high electro-chemo-mechanical activity based on glucose-gelatin nanofiber scaffolds coated with polypyrrole and doped with dodecylbenzensulfonate or with triflouromethanesulfonate. The experiments carried out in the manuscript are described in sufficient detail. The composition, electronic , ionic conductivity and the electro-chemo-mechanical activity was characterized by means of different characterization techniques. In my opinion, this manuscript is very interesting and the results, in addition to be well organized and discussed, they were compared and contrasted with literature data, being the conclusions presented supported by the reported data. Therefore, in my opinion, the manuscript is acceptable for publication as it is.

We thank the reviewer for supporting our work and the positive comments

Reviewer #3: I have reviewed the research manuscript draft "Concept of an artificial muscle design on polypyrrole nanofiber scaffolds" written by Madis Harjo, Martin Järvekülg, Tarmo Tamm, Toribio F. Otero and Rudolf Kiefer submitted for publication in PLOS ONE.

We thank the reviewer for his positive review and have correct our mistakes in the revised manuscript.

After the reading of the manuscript, in my opinion it is an interesting work where the obtained results are reasonable well explained and could gives rise to a future research works in this line

Acceptable under minor revisions. Only I have few comments or suggestions.

We thank the reviewer for finding the mistakes and suggestions to improve the manuscript.

1: Replace in the manuscript and supplementary information:

Madis Harjo1, Martin Järvekülg2, Tarmo Tamm1, Toribio F. Otero3, and Rudolf Kiefer4,*

by

Madis Harjo1, Martin Järvekülg2, Tarmo Tamm1, Toribio F. Otero3 and Rudolf Kiefer4*

Corrected.

2: In the abstract replace: triflouromethanesulfonate by trifluoromethanesulfonate

We have corrected this mistake in abstract.

3: In the introduction, at the end of second paragraph: Comparable, without capital letter: comparable

The term is corrected.

4: Section 2.2: in the first (40ºC) and second paragraph (60ºC, 60ºC) replace de cero in superindex by degree symbol. 

We have replaced the superscript degree symbol.

5: At the end of equation 3, 4 and 5, remove the comma, point and comma, respectively

We have corrected equation 3-5 and removed comma and points

6: Section 3.2.1, in first line of second paragraph: Figure 3b instead Figure 2b

We thank the reviewer pointing this out and have corrected it

7: Last line before section 3.2.2: “cycles ,”  “cycles,”

We have corrected this.

8: Section 3.2.2 in first paragraph: “CFS-PPy/DBS” instead of “PPy/DBS”

The term is corrected.

9: Section 3.2.2 in the paragraph prior to Figure 5:

Revise the sentence “Figure 5a shows the linear dependence of the diffusion coefficients during the oxidation reaction (S2 Table) on the applied frequency (S3 Figure) depicts the evolution during reduction reactions)”

We agree and have reformulated

Figure 5a shows the linear dependence of the diffusion coefficients during the oxidation reaction on the applied frequency. The diffusion coefficients upon reduction against the applied frequency are shown at S3 Figure.

10: Section 3.2.2 first paragraph after Figure 5: “sentence” instead of “sentiency”.

 We thank the reviewer pointing that out and have corrected it.

---

## [Editor Report · Decision Letter 1]

23 Apr 2020

Concept of an artificial muscle design on polypyrrole nanofiber scaffolds

PONE-D-20-03013R1

Dear Dr. Kiefer,

We are pleased to inform you that your manuscript has been judged scientifically suitable for publication and will be formally accepted for publication once it complies with all outstanding technical requirements.

With kind regards,

Susana Cristina de Matos Fernandes

Academic Editor

PLOS ONE
---

## [Editor Report · Acceptance letter]

28 Apr 2020

PONE-D-20-03013R1 

Concept of an artificial muscle design on polypyrrole nanofiber scaffolds 

Dear Dr. Kiefer:

I am pleased to inform you that your manuscript has been deemed suitable for publication in PLOS ONE. Congratulations! Your manuscript is now with our production department. 

With kind regards,

on behalf of

Dr. Susana Cristina de Matos Fernandes 

Academic Editor

PLOS ONE